# Alleviation of Immobilization Stress or Fecal Microbiota-Induced Insomnia and Depression-like Behaviors in Mice by *Lactobacillus plantarum* and Its Supplement

**DOI:** 10.3390/nu16213711

**Published:** 2024-10-30

**Authors:** Dong-Yun Lee, Ji-Su Baek, Yoon-Jung Shin, Dong-Hyun Kim

**Affiliations:** 1Neurobiota Research Center, College of Pharmacy, Kyung Hee University, 26, Kyungheedae-ro, Dongdaemun-Gu, Seoul 02447, Republic of Korea; dongyun8246@naver.com (D.-Y.L.); bjisoo500@naver.com (J.-S.B.); nayo971111@naver.com (Y.-J.S.); 2PBLbioLab, Inc., Seoul 03174, Republic of Korea

**Keywords:** stress, depression, sleep disturbance, *Lactobacillus plantarum*, hempseed oil, neuroinflammation

## Abstract

Insomnia (sleeplessness) is a potential symptom of stress-induced depression/anxiety (DA), which induces TNF-α expression. Therefore, this study aimed to examine the effect of *Lactobacillus* (*Lactiplantibacillus*) *plantarum* P72, isolated as a strain suppressing lipopolysaccharide-induced expression of TNF-α in Caco2 cells, on DA and insomnia in immobilization stress (IS)- or cultured fecal microbiota (cFM)-treated mice. Oral administration of live or heat-killed P72 (hP72) reduced IS- or cFM-induced DA-like behaviors. They also reduced sleep latency time (SLT) and enhanced sleep duration (SLD). Additionally, P72 upregulated γ-aminobutyric acid (GABA), GABA_A_ receptor α1, serotonin, and 5-HT_1A_ receptor expression, which were downregulated by IS or cFM. Hempseed oil (HO) alone was ineffective against IS-induced DA- and insomnia-like behaviors, but its combination with P72 (PH) or hP72 (hPH) showed enhanced efficacy, reducing DA- and insomnia-like behaviors more strongly than P72 or HO alone. These also reduced the number of NF-κB-positive cells and the expression of TNF-α in the prefrontal cortex and colon. These results imply that P72 and its combination with HO can alleviate DA and insomnia by upregulating serotonergic and GABAergic systems through the suppression of NF-κB signaling.

## 1. Introduction

Chronic exposure to stress, including social defeat and immobilization, triggers the release of adrenal hormones like noradrenaline, adrenaline, and glucocorticoids, as well as proinflammatory cytokines like TNF-α and IL-6, leading to depression and anxiety (DA) [1,2]. Stress-induced DA fluctuates the release of serotonin (5-HT) and γ-aminobutyric acid (GABA) from enterochromaffin and neural cells [3,4]. DA triggers sleep disturbance (SD): the majority of patients with DA suffer from insomnia [5]. Therefore, antidepressants and serotonin inducers are often used to treat both DA and insomnia [3,6].

Many *lactobacilli* and *bifidobacteria* are known to exhibit beneficial activities against immune and psychiatric disorders by regulating the immune, endocrine, and neural systems [7,8,9]. *Lactobacillus plantarum* D-9, *Lactobacillus reuteri* NK33, *Bifidobacterium longum* 1714, *B. longum* CCFM687, and *Bifidobacterium adolescentis* NK98 alleviate DA-like behavior in stress-induced rodents [10,11,12,13]. *L. plantarum* JYLP-326 and *Bifidobacterium breve* CCFM1025 improve stress-induced sleep disorder in mice [14,15]. *L. plantarum* P8 alleviates anxiety in stressed adults [16]. *B. longum* reduces the psychological stress score in healthy adults [17]. *B. longum* NCC3001 decreases DA scores in patients with irritable bowel syndrome [18]. *L. plantarum* PS128 attenuates DA-related symptoms in volunteers with self-reported insomnia [19]. NVP1704 (an NK33 and NK98 mix) also alleviates stress-induced DA in mice and depressive symptoms and sleeplessness in healthy adults [20]. Nevertheless, the action mechanism of probiotics against sleep disorders including insomnia is not fully understood.

Therefore, we isolated *Lactobacillus* (*Lactiplantibacillus*) *plantarum* P72, which increased the secretion of serotonin in vitro, from the bacterial collection of healthy human feces and examined the effect of P72 (live), heat-killed P72 (hP72), and their combination with hempseed oil (HO) on immobilization stress (IS)- or cultured fecal microbiota (cFM)-induced depression and SD (insomnia) in mice.

## 2. Materials and Methods

### 2.1. Culture of P72

P72 (KCCM13445P, deposited in the Korean Culture Center for Microorganisms) was cultured in general media such as De Man—Rogosa—Sharpe (MRS) broth and centrifuged (5000× *g*, 4 °C, and 20 min). The precipitate was suspended in distilled water (for the in vitro experiment) or 0.1% trehalose (for the in vivo experiment) and freeze-dried. The number of live P72 was determined using plate counting on MRS agar plates. Heat-killed P72 (hP72) was prepared by heating P72 suspended in distilled water (for the in vitro experiment) or 1% trehalose (for the in vivo experiment) at 80 °C for 15 min and freeze-drying.

### 2.2. Culture of SH-SY5Y and Caco-2 Cells

SH-SY5Y or Caco-2 cells were cultured as previously reported [20]. SH-SY5Y cells (1 × 10^6^ cells/mL) were incubated with corticosterone (300 μM, Sigma [20]) in the absence or presence of probiotics (1 × 10^5^ CFU/mL) for 24 h. In the supernatant of the culture, the serotonin level was assayed using its assay kit (DLD Diagnostika GmbH, Hamburg, Germany). Caco-2 cells (1 × 10^6^ cells/mL) were incubated with lipopolysaccharide (LPS, 100 ng/mL, Sigma [20]) in the presence or absence of probiotics (1 × 10^5^ CFU/mL) for 24 h. The TNF-α concentration was determined using its enzyme-linked immunosorbent assay (ELISA) kit (R&D system).

### 2.3. Animals

C57BL/6 mice (male, 18–21 g, 6 weeks old) were obtained from Koatech Co., Ltd. (PyungTaek-shi, Republic of Korea) and kept in plastic cages with a 5 cm raised wire floor under controlled conditions, as previously reported [20]. All animal experiments were ethically approved by the Committee for the Care and Use of Laboratory Animals in Kyung Hee University (IACC, KHUASP(SE)-23545, 14 March 2023) and were conducted according to the Ethical Policies and Guidelines of the University for Laboratory Animals Care and Use, and the Use of Laboratory Animals and ARRIVE guidelines [21].

### 2.4. Induction of DA- and Insomina-like Symptom (DI_L_S) in Mice

Experiment 1—Mice were divided into 3 groups (NC, IS, and P72) consisting of 8 mice. IS and P72 groups were exposed to IS daily for 5 days [13]. Test agents (IS, saline; P72, 1 × 10^9^ CFU/mouse/day of P72) were orally administered daily for 5 days from next day after the final IS treatment. The IS-untreated normal control (NC) group was gavaged with saline.

Experiment 2—Mice were divided into 3 groups (NC, cFM, and P72) consisting of 6 mice. The cultured fecal microbiota (2 × 10^8^ colony-forming unit [CFU]/mouse, suspended in 0.1 mL of saline; the number of cultured fecal microbiota of patients with depression and inflammatory bowel disease was counted using general anaerobic medium (Nissui Pharm. Co., Tokyo, Japan)). The culture fecal microbiota, which were prepared as previously reported [22], were orally gavaged in cFM and P72 groups daily for 5 days. Thereafter, test agents (cFM, saline; P72, 1 × 10^9^ CFU/mouse/day of P72) were orally administered daily for 7 days. NC was treated with saline.

Experiment 3—Mice were divided into 7 groups (NC, IS, P72_L_, P72_H_, HO_H_, PH, and DP) consisting of 8 mice. Mice (IS, P72_L_, P72_H_, HO_H_, PH, and DP) were treated with IS daily for 5 days. Thereafter, test agents (IS, saline; P72_L_, 0.4 × 10^9^ CFU/mouse/day of P72; P72_H_, 1 × 10^9^ CFU/mouse/day of P72; HO_H_, 0.24 g/kg of HO; PH, 0.4 × 10^9^ CFU/mouse/day of P72 and 0.12 g/kg of HO; and DP, 20 mg/kg of diphenhydramine) were orally administered (for P72 and HO) or intraperitoneally (for diphenhydramine) daily for 7 days. NC was treated with saline.

Experiment 4—Mice were separated into 3 groups (NC, IS, hP72, hPH) consisting of 6 mice. Mice (IS, hP72, and hPH) were treated with IS daily for 5 days. Thereafter, test agents (IS, vehicle alone; hP72, 1 × 10^9^ CFU/mouse/day of hP72; hPH, 1 × 10^9^ CFU/mouse/day of hP72 and 0.12 g/kg of HO) were orally administered daily for 7 days. NC was treated with saline alone.

DA-like behaviors (total distance moved (TD), distance moved in the center area (DC) and time moved in the center area (TC) in the open field test (OFT), time moved in open arms (OT) and entries in open arms (OE) in the elevated plus maze task (EPMT), and immobility time (IT) in the tail suspension test (TST)) were measured (between 1:00 and 5:00 p.m.), as previously reported [20]. On next day (between 1:00 and 5:00 p.m.), sleep latency time (SLT) and sleep duration (SLD) were measured after the peritoneal injection of pentobarbital sodium (PS) or exposure to isoflurane (IF) saturated in the box [23]. Detailed protocols are described in the Appendix A.

Mice were euthanatized by exposure to CO_2_ in a plastic box 24 h after the end of the final experiment, and cervically dislocated. Brain and colon tissues were collected and kept in the freezer (−80 °C) for the measurement of biomarkers. For the immunofluorescence staining, mice were transcardially perfused, as previously reported [13].

### 2.5. ELISA, Quantitative Polymerase Chain Reaction (qPCR), and Immunofluorescence Staining

The expression levels of inflammatory biomarkers in the brain (prefrontal cortex) and colon tissues were assayed using ELISA [20]. Serotonin 1A receptor (5-HT_1A_R), 5-HT_1B_R, melatonin receptor type 1 (MT1R), MT2R, GABA type A receptor subunit alpha1 (GABA_A_Rα1), GABA_A_Rα2, and glyceraldehyde-3-phosphate dehydrogenase (GAPDH) expression levels were assayed using qPCR. Primers are indicated in Appendix A. Their tissue sections were immunofluorescence-stained, as previously reported [13,20]. Detailed methods are indicated in the Appendix A.

### 2.6. Whole Genome Analysis

The sequencing libraries for the P72 whole genome analysis were prepared, as previously reported [24], and the P72 genome sequence (6 contigs) was obtained.

### 2.7. Statistical Analysis

Experimental data are described as mean ± standard deviation (SD) and the significance was analyzed by a one-way ANOVA followed by Tukey’s multiple comparison test (*p* < 0.05), using a GraphPad Prism 9. The resulting F-value and degrees of freedom for datasets of each experiment were discussed.

## 3. Results

### 3.1. P72 Increased the Secretion of Serotonin in Corticosterone-Treated SH-SY5Y Cells

First, we screened probiotics inducing serotonin secretion in corticosterone-treated SH-SY5Y cells from the bacterial collection of healthy human feces. Among tested lactobacilli, P72 potently enhanced corticosterone-suppressed serotonin release (Figure 1a). P72 also suppressed TNF-α expression in LPS-stimulated Caco-2 cells (Figure 1b). P72 was identified as *Lactiplantibacillus plantarum*, based on Gram staining, API kit, and 16S rDNA and whole genome sequencing. The genome of P72 was 3,350,919 base pairs (contigs 6) with a GC content of 44.4%. The total number of CDS was 3135 (Appendix A). The tRNA gene number was 70. The rRNA gene number was 16. The P72 genome sequence showed a phylogenetic similarity to *Lactiplantibacillus plantarum* NCTC13644 (99.15%) using OrthoANI.

### 3.2. P72 Mitigated DI_L_S in IS-Exposed Mice

To comprehend whether P72 could improve DA and insomnia, we investigated its impact in mice exposed to IS. IS treatment increased DA-like behavior: it decreased TD, DC, and TC in the OFT to 85.3% (F_2,21_ =18.3, *p* < 0.001), 55.8% (F_2,21_ = 17.6, *p* < 0.001), and 41.0% (F_2,21_ = 27.2, *p* < 0.001) of NC, respectively, decreased OT and OE in the EPMT to 45.6% (F_2,21_ = 15.7, *p* < 0.001) and 41.4% (F_2,21_ = 38.4, *p* < 0.001) of NC, respectively, and increased IT in the TST to 142.8% (F_2,21_ = 60.4, *p* < 0.001) of NC (Figure 2a–g). However, orally administered P72 (1 × 10^9^ CFU/mouse/day) significantly recovered IS-decreased TD, DC, and TC to 108.6%, 99.4%, and 92.4% of NC, respectively, IS-suppressed OT and OE to 119.2% and 104.2% of NC, respectively, and IS-increased IT to 112.5% of NC.

Exposed IS caused insomnia in mice, increasing SLT to 126.0% (F_2,21_ = 23.0, *p* < 0.001) of NC and decreasing SLD to 83.0% (F_2,21_ = 4.5, *p* = 0.023) of NC (Figure 2h,i). However, P72 significantly improved insomnia: it decreased IS-increased SLT to 96.4% of NC and enhanced IS-shortened SLD to 101.3% of NC.

IS exposure significantly downregulated the expression of GABA, its receptors GABA_A_Rα1 and GABA_A_Rα2, serotonin, and its receptors 5-HT_1A_R and 5-HT_1B_R in the prefrontal cortex (Figure 3a–f). Furthermore, IS increased corticosterone and TNF-α levels, while decreasing IL-10 expression (Figure 3g–i). Orally administered P72 upregulated IS-decreased GABA, GABA_A_Rα1, GABA_A_Rα2, serotonin, 5-HT_1A_R, and 5-HT_1B_R expression levels. P72 also decreased IS-induced corticosterone and TNF-α levels, while increasing IL-10 levels.

### 3.3. P72 Alleviated Colitis in IS-Exposed Mice

Exposed IS caused colitis in mice, decreasing colon length and IL-10 levels and increasing myeloperoxidase, TNF-α, IL-1β, and IL-6 levels in the colon (Figure 4). Orally administered P72 (1 × 10^9^ CFU/mouse/day) significantly increased IS-shortened colon length and downregulated IS-induced myeloperoxidase, TNF-α, IL-1β, and IL-6 levels, while upregulating IL-10 levels.

### 3.4. P72 Alleviated cFM Transplantation-Induced DI_L_S and Gut Inflammation in Mice

Next, we investigated the effect of P72 on cFM transplantation-induced DA- and insomnia-like behavior in mice. cFM transplantation reduced TD, DC, and TC in OFT to 86.3% (F_2,21_ = 13.5, *p* < 0.01), 63.3% (F_2,21_ = 33.8, *p* < 0.01), and 49.0% (F_2,21_ = 48.0, *p* < 0.01) of NC, respectively, and OT and OE in EPMT to 54.3% (F_2,21_ = 14.4, *p* = 0.01) and 51.3% (F_2,21_ = 49.1, *p* < 0.01) of NC, respectively (Figure 5a–f). cFM transplantation also increased IT in TST to 124.7% (F_2,21_ = 44.7, *p* < 0.01) of NC (Figure 5g). However, orally administered P72 (1 × 10^9^ CFU/mouse/day) significantly recovered cFM-decreased TD, DC, and TC to 97.2%, 90.7%, and 70.8% of NC, respectively, and OT and OE to 120.0% and 112.2% of NC, respectively. P72 also restored cFM-increased IT to 77.5% of NC.

cFM transplantation increased SLT to 119.5% (F_2,21_ = 18.1, *p* < 0.001) of NC and reduced SLD to 50.9% (F_2,21_ = 61.2, *p* < 0.001) of NC (Figure 5h,i). However, orally administered P72 significantly recovered cFM-increased SLT to 105.5% of NC and raised cFM-decreased SLD to 73.2% of NC.

cFM transplantation decreased GABA, GABA_A_Rα1, GABA_A_Rα2, serotonin, 5-HT_1A_R, and 5-HT_1B_R expression in the prefrontal cortex (Figure 6a–f). cFM transplantation increased corticosterone and TNF-α levels, while decreasing IL-10 levels (Figure 6g–i). Orally administered P72 upregulated cFM transplantation-suppressed GABA, GABA_A_Rα1, GABA_A_Rα2, serotonin, 5-HT_1A_R, and 5-HT_1B_R expression. P72 also decreased cFM transplantation-induced corticosterone and TNF-α expression, while increasing cFM transplantation-suppressed IL-10 expression.

cFM transplantation caused inflammation in the colon of mice: it shortened colon length, increased myeloperoxidase, TNF-α, IL-1β, and IL-6 levels, and decreased IL-10 levels (Figure 7). Orally administered P72 significantly recovered suppressed cFM transplantation-increased myeloperoxidase, TNF-α, IL-1β, and IL-6 levels.

### 3.5. P72 and Its Combination with HO (PH) Alleviated DI_L_S and Gut Inflammation in IS-Exposed Mice

The effect of P72 combined with HO (PH) on DA- and insomnia-like behavior was investigated in IS-exposed mice (Figure 8). Exposure to IS significantly reduced TD, DC, and TC in OFT to 88.8% (F_6,49_ = 9.0, *p* < 0.01), 69.5% (F_6,49_ = 13.0, *p* < 0.01), and 46.8% (F_6,49_ = 17.7, *p* < 0.01) of NC, respectively, and OT and OE in EPMT to 37.5% (F_6,49_ = 25.6, *p* < 0.01) and 49.9% (F_6,49_ = 30.4, *p* < 0.01) of NC, respectively (Figure 8a–f). IS increased IT in TST to 143.1% (F_6,49_ = 33.1, *p* < 0.01) of NC (Figure 8g). However, orally administered P72 and PH (0.4 × 10^9^ and 1 × 10^9^ CFU/mouse/day) significantly increased IS-suppressed TD, DC, and TC: P72 at 0.4 × 10^9^ CFU/mouse (P72_L_) and PH (0.4 × 10^9^ CFU/mouse of P72 + 0.12 g/kg of HO [HO_L_]) increased TD to 94.8% and 103.2%, respectively, DC to 93.3% and 97.0% of NC, respectively, and TC to 91.8% and 95.3%, respectively. However, HO at 0.24 g/kg (HO_H_) did not affect TD, DC, or TC. P72_L_, PH, and HO_H_ increased IS-induced OT in EPMT to 114.9%, 137.8%, and 94.9% of NC, respectively, and OT to 79.2%, 73.42%, and 85.4% of NC, respectively. P72_L_ and PH reduced IT to 105.8% and 110.5% of NC. However, HO did not affect IT.

Exposed IS enhanced insomnia-like behavior in mice: it increased SLT to 131.3% (F_6,49_ = 28.7, *p* < 0.01) of NC and reduced SLD to 44.5% (F_6,49_ = 39.0, *p* < 0.01) of NC (Figure 8h,i). However, P72_L_ and PH significantly reduced IS-increased SLT to 116.7% and 111.2% of NC, respectively. However, HO_H_ did not significantly affect IS-increased SLT. P72_L_, PH, and HO_H_ recovered IS-decreased SLD to 75.3%, 95.2% and 64.0% of NC, respectively.

IS exposure downregulated the expression of GABA, GABA_A_Rα1, GABA_A_Rα2, MT1R, MT2R, serotonin, 5-HT_1A_R, 5-HT_1B_R, and the number of GABA_1A_Rα1-positive cells (Figure 9). Orally administered P72_L_, P72_H_, or PH upregulated IS-suppressed GABA, serotonin, GABA_A_Rα1 and GABA_A_Rα2, MT1R, MT2R, 5-HT_1A_R, and 5-HT_1B_R expression and GABA_1A_Rα1-positive cell numbers. Diphenhydramine did not induce IS-decreased levels of GABA, serotonin, and their receptors except GABA_1A_Rα2 and MT1R.

Exposed IS upregulated corticosterone, TNF-α, and IL-6 levels and NF-κB^+^ Iba1^+^ cell numbers, and decreased IL-10 levels in the prefrontal cortex (Figure 10). Orally administered P72_L_, P72_H_, or PH significantly reduced IS-increased biomarker levels, while increasing IS-decreased IL-10 levels.

Exposure to IS caused inflammation in the colon of mice: it decreased colon length and IL-10 levels and increased myeloperoxidase, TNF-α, IL-1β, and IL-6 levels and NF-κB^+^ CD11c^+^ cell populations (Figure 11). Oral administration of PH and P72 significantly recovered IS-increased myeloperoxidase, TNF-α, IL-1β, and IL-6 levels and NF-κB^+^ CD11c^+^ cell populations and IS-decreased IL-10 levels.

### 3.6. hP72 and Its Combination with HO (hPH) Alleviated IS-Induced DI_L_S in Mice

The effects of hP72 and hPH against DI_L_S was investigated in mice exposed to IS. Exposed IS reduced DC and TC in OFT to 49.5% (F_3,28_ = 11.6, *p* < 0.001) and 40.7% (F_3,28_ = 9.2, *p* < 0.001) of NC, respectively, and OT in EPMT to 43.21% (F_3,28_ = 10.4, *p* < 0.001) of NC, while increasing IT in TST to 132.8% (F_3,28_ = 12.22, *p* < 0.001) of NC (Figure 12a–e). Orally administered hP72 or hPH significantly increased IS-suppressed DC to 88.9% and 96.9% of NC, respectively, TC to 74.8% and 89.1% of NC, respectively, and OT to 74.7% and 85.1% of NC, respectively. They also decreased IT to 115.6% and 100.2% of NC, respectively.

Exposed IS enhanced insomnia-like behavior in mice: it increased SLT to 186.5% (F_3,28_ = 12.58, *p* < 0.001) of NC and reduced SLD to 53.6% (F_3,28_ = 32.75, *p* < 0.001) of NC (Figure 12f,g). Orally administered hP72 or hPH significantly reduced IS-increased SLT to 105.5% and 111.5% of NC, respectively, and enhanced IS-decreased SLD to 69.9% and 87.2% of NC, respectively. hP72 and hPH significantly increased IS-suppressed GABA and serotonin levels in the prefrontal cortex. They downregulated IS-increased corticosterone and TNF-α levels, while increasing IS-decreased IL-10 expression.

## 4. Discussion

We found that IS exposure and cFM transplantation increased DA- and sleep disturbance-like behaviors in mice, suppressing GABA and serotonin levels and inducing TNF-α levels and NF-κB-positive cell populations in the brain. Stress-induced DA triggers sleep disturbance, including insomnia and systemic inflammation [1,25,26]. Systemic inflammation including gut inflammation induces DA [27,28]. Depression has been suggested to be closely associated with reduced serotonin and GABA concentrations and increased TNF-α concentrations, which suppresses the release of serotonin and GABA, although there are many conflicting results [29,30,31]. Stress also causes gut dysbiosis, which is strongly involved with the outbreak of DA and systemic inflammation [32]. Fecal microbiota transplantation from DA patients induces DA-like behavior and neuroinflammation in transplanted mice [33]. These observations imply that depression-induced gut dysbiosis can induce sleep disturbance, including insomnia, by decreasing GABA and serotonin levels and increasing NF-κB-mediated TNF-α expression.

P72 downregulated LPS-increased TNF-α levels in Caco-2 cells and upregulated LPS-decreased serotonin secretion in SH-SY5Y cells. P72 also upregulated GABA and serotonin concentrations and downregulated TNF-α and IL-6 levels and NF-κB-positive cell numbers in IS- or cFM-exposed mice. P72 decreased IS-induced DA- and sleep disturbance-like behavior. *Lactobacillus mucosae* NK41 alleviates depression in mice [34]. Anti-inflammatory *Bifidobacterium infantis* CCFM687 alleviates depression in mice [12]. Anti-inflammatory *Lactobacillus plantarum* NK151 also mitigates depression in mice [22]. These observations imply that P72 can mitigate depression and sleep disturbance by inducing GABA and serotonin production and suppressing NF-κB-mediated TNF-α expression.

P72 increased the levels of GABA and serotonin and the expression of GABA receptors GABA_A_Rα1 and GABA_A_Rα2, 5-HT receptors 5-HT_1A_R and 5-HT_1B_R, and melatonin receptors MT1R and MT1R. GABA and its receptor agonists such as benzodiazepine also alleviates DA and insomnia [35]. 5-HT_1A_R and HT_1B_R agonists, such as tandospirone, mitigate depression and insomnia [36,37,38]. Diazepam, a positive allosteric modulator of the GABA_A_ receptor, improves sleep disturbance (including insomnia) by increasing GABA_A_R [39]. Ramelteon, a melatonin receptor agonist, also improves DA and insomnia symptoms [29]. Stress upregulates TNF-α expression in immune cells, which downregulates the expression and actions of serotonin, GABA, and their receptors [40,41]. P72 suppressed TNF-α expression and NF-κB activation in vitro and in vivo. These observations imply that P72 can alleviate DA and sleep disturbance, including insomnia, by regulating serotonergic and GABA_A_ergic systems through the suppression of NF-κB signaling.

hP72, which was heat-killed, also alleviated DA and sleep disturbance in mice, while HO did not. Nevertheless, PH (a mix of P72 and HO) and hPH (a mix of hP72 and HO) alleviated DA and sleep disturbance, including insomnia, more potently than P72 or hP72 alone. They increased GABA, serotonin, and their receptor expression and decreased TNF-α expression. These observations imply that the active components of P72 against DA and sleep disturbance may be heat-stable, and their action mechanism may exhibit a significant difference.

## 5. Conclusions

P72 (live) and hP72 (heat-killed) upregulated 5-HT, GABA, and their receptor levels, and downregulated corticosterone and TNF-α levels and NF-κB activation in the brain and intestine. P72 decreased DA- and insomnia-like behavior. PH more strongly alleviated DI_L_S than P72 or HO alone. The efficacies of hP72 and hPH were significantly different to those of P72 and PH, respectively. Finally, P72 and its supplement PH may mitigate DA and insomnia by upregulating serotonergic and GABA_A_-ergic systems through the suppression of NF-κB signaling.

## Figures and Tables

**Figure 1 nutrients-16-03711-f001:**
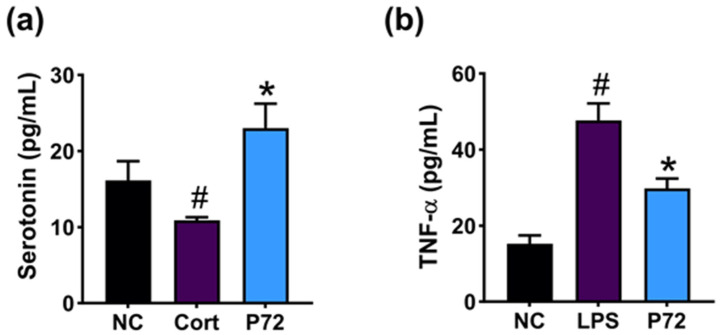
Effect of P72 on the release of corticosterone-suppressed serotonin in SH-SY5Y cells (**a**) and the expression of LPS-induced TNF-α in Caco-2 cells (**b**). In SH-SY5Y cells, NC, saline; Cort, 300 μM corticosterone; P72, 1 × 10^5^ CFU/mL of P72 + corticosterone. In Caco2 cells, NC, saline; LPS, 100 ng/mL of LPS; P72, 1 × 10^5^ CFU/mL of P72 + LPS. Each group represents the mean ± SD (*n* = 4). ^#^ *p* < 0.05 vs. NC. * *p* < 0.05 vs. Cort/LPS.

**Figure 2 nutrients-16-03711-f002:**
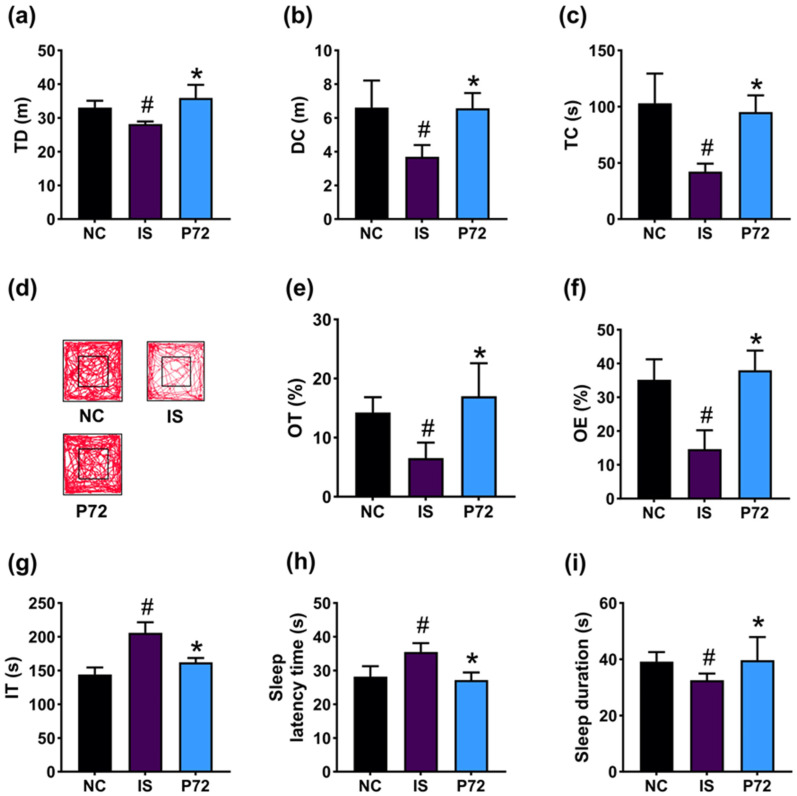
Effect of P72 on IS-induced DA- and insomnia-like behaviors in mice. Effect of P72 on TD (**a**), DC (**b**), TC (**c**), and travel pathway (**d**) in OFT. Effect on OT (**e**) and OE (**f**) in the EPMT and IT in TST (**g**). Effect on SLT (**h**) and SLD (**i**). IS, saline; P72, 1 × 10^9^ CFU/mouse/day of P72 in IS/IF-treated mice; NC, saline in mice (not exposed IS/IF). Data values indicate mean ± SD (*n* = 8). ^#^ *p* < 0.05 vs. NC. * *p* < 0.05 vs. IS.

**Figure 3 nutrients-16-03711-f003:**
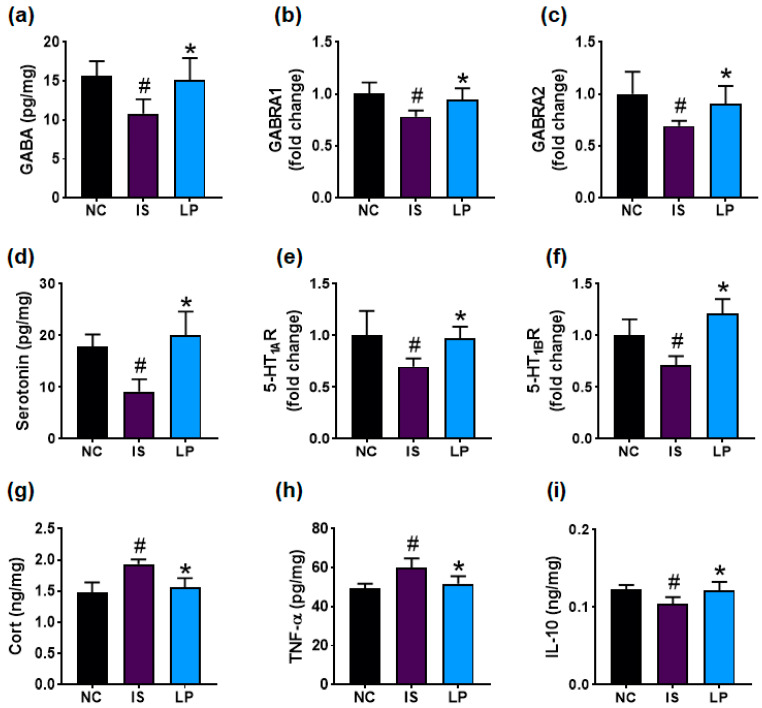
Effect of P72 on the expression of DA- and insomnia-related markers in the prefrontal cortex. Effect on GABA (**a**), GABA_A_Rα1 (**b**), and GABA_A_Rα2 (**c**) expression levels. Effect on serotonin (**d**), 5-HT_1A_R (**e**), and 5-HT_1B_R (**f**) expression levels. Effect on corticosterone (Cort, (**g**)), TNF-α (**h**), and IL-10 (**i**) levels. IS, vehicle; P72, 1 × 10^9^ CFU/mouse/day of P72 in IS/IF-treated mice; NC, saline in IS/IF-nontreated mice. Data are mean ± SD (n = 8). ^#^ *p* < 0.05 vs. NC. * *p* < 0.05 vs. IS.

**Figure 4 nutrients-16-03711-f004:**
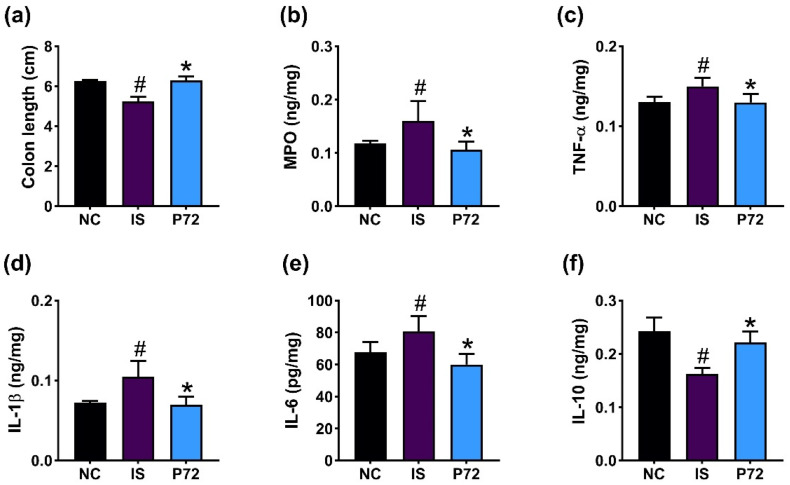
Effect of P72 on IS-induced colitis in mice. Effect on colon length (**a**) and myeloperoxidase (MPO, (**b**)), TNF-α (**c**), IL-1β (**d**), IL-6 (**e**), and IL-10 levels (**f**) in the colon. IS, vehicle; P72, 1 × 10^9^ CFU/mouse/day of P72 in IS/IF-treated mice; NC, saline in IS/IF-untreated mice. Data are mean ± SD (*n* = 8). ^#^ *p* < 0.05 vs. NC. * *p* < 0.05 vs. IS.

**Figure 5 nutrients-16-03711-f005:**
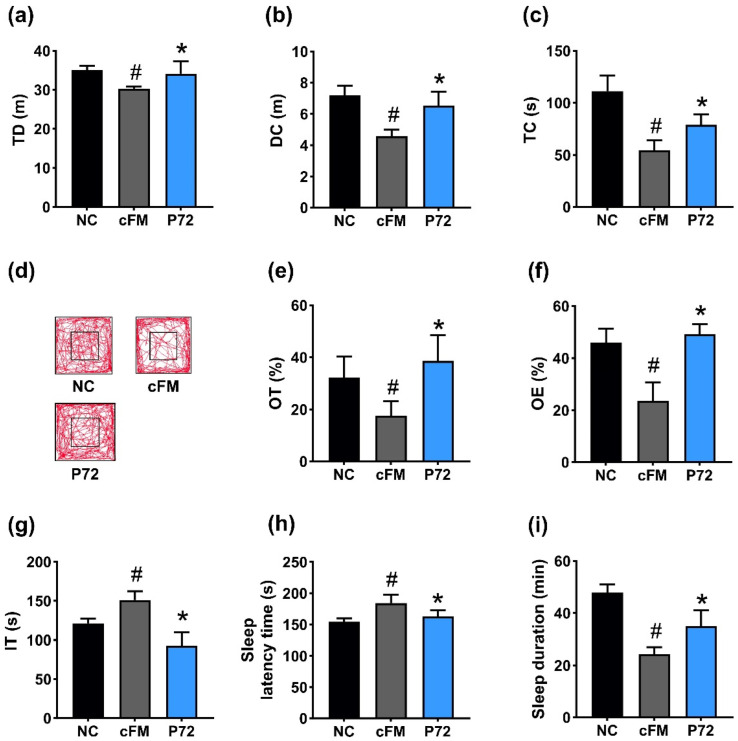
Effect of P72 on DA- and insomnia-like behavior in cFM-transplanted mice. Effect of P72 on TD (**a**), DC (**b**), TC (**c**), and travel pathway (**d**) in OFT. Effect on OT (**e**) and OE (**f**) in EPMT and IT in TST (**g**). Effect on SLT (**h**) and SLD (**i**). IS, vehicle; P72, 1 × 10^9^ CFU/mouse/day of P72 in cFM/IF-treated mice; NC, vehicle (saline) in cFM/IF-nontreated mice. Data values indicate mean ± SD (*n* = 6). ^#^ *p* < 0.05 vs. NC. * *p* < 0.05 vs. cFM.

**Figure 6 nutrients-16-03711-f006:**
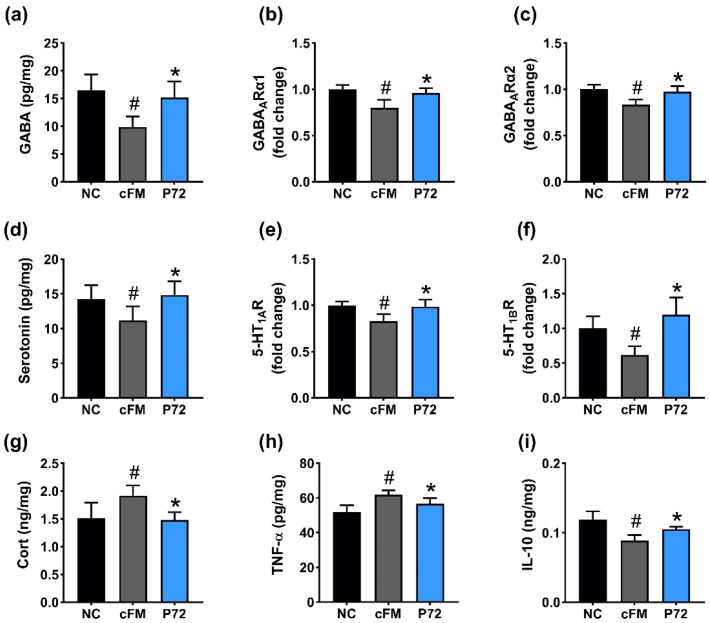
Effect of P72 on cFM transplantation-induced DA- and insomnia-related markers in the prefrontal cortex. Effect of P72 on GABA (**a**), GABA_A_Rα1 (**b**), and GABA_A_Rα1 (**c**) levels. Effect on serotonin (**d**), 5-HT_1A_R (**e**), and 5-HT_1B_R (**f**) levels. Effect on corticosterone (**g**), TNF-α (**h**), and IL-10 (**i**) levels. cFM, vehicle; P72, 1 × 10^9^ CFU/mouse/day of P72 in cFM/IF-treated mice; NC, vehicle (saline) in cFM/IF-untreated mice. Data are mean ± SD (*n* = 6). ^#^ *p* < 0.05 vs. NC. * *p* < 0.05 vs. cFM.

**Figure 7 nutrients-16-03711-f007:**
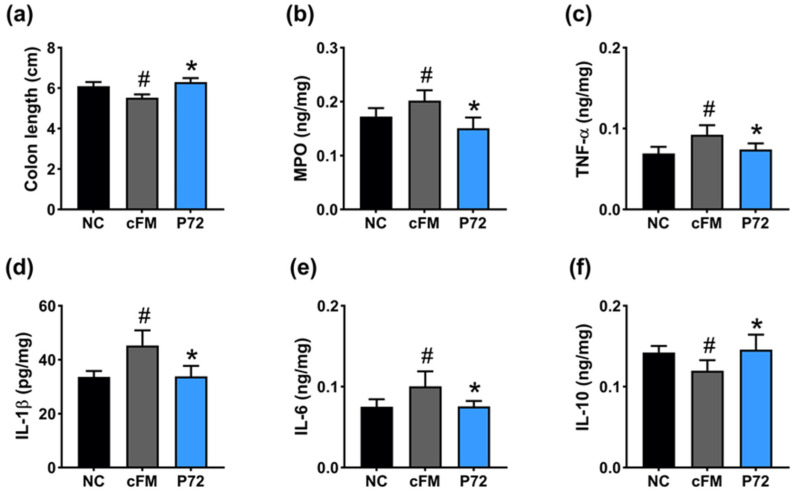
Effect of P72 on cFM transplantation-induced colitis in mice. Effect on colon length (**a**), myeloperoxidase (MPO, (**b**)), TNF-α (**c**), IL-1β (**d**), IL-6 (**e**), and IL-10 expression (**f**) in the colon. cFM, vehicle; P72, 1 × 10^9^ CFU/mouse/day of P72 in cFM/IF-treated mice; NC, vehicle (saline) in cFM/IF-nontreated mice. Data are mean ± SD (*n* = 6). ^#^ *p* < 0.05 vs. NC. * *p* < 0.05 vs. IS.

**Figure 8 nutrients-16-03711-f008:**
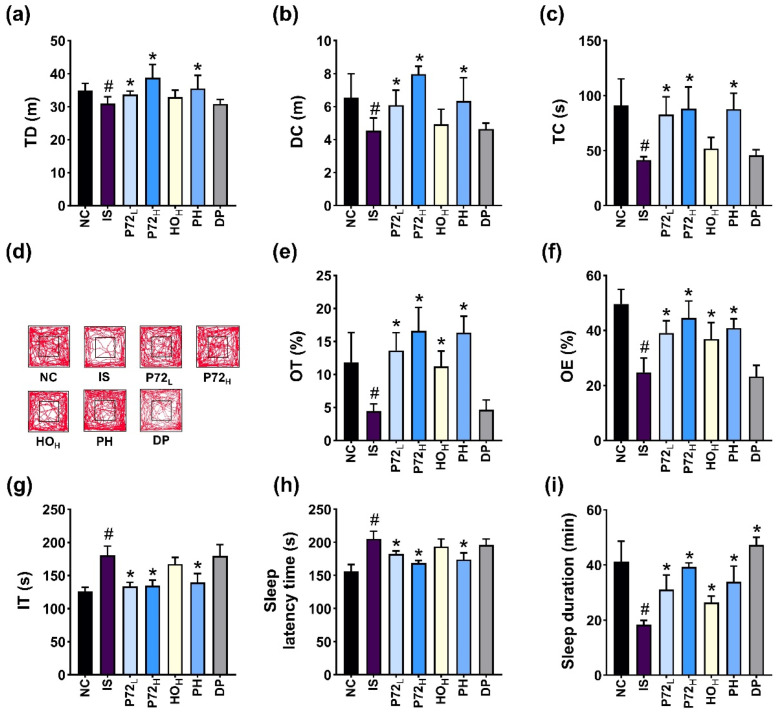
Effects of P72, HO, and PH on IS-induced DA- and insomnia-like behavior in mice. Effects on OT (**a**) and OE (**b**) in EPMT and IT in TST (**c**). Effects on TD (**d**), CD (**e**), TC (**f**), and travel pathway (**g**) in OFT. Effects on SLT (**h**) and SLD (**i**). IS, vehicle; P72_L_, 4 × 10^8^ CFU/mL of P72; P72_H_, 1 × 10^9^ CFU/mouse of P72; HO_H_, 0.24 g/kg of HO; PH, 0.4 × 10^9^ CFU/mouse/day of P72 and 0.12 g/kg of HO; DP, 20 mg/kg DPH, in IS/PS-exposed mice; NC, vehicle, in IS/PS-nontreated mice. Data are mean ± SD (*n* = 8). ^#^ *p* < 0.05 vs. NC. * *p* < 0.05 vs. IS.

**Figure 9 nutrients-16-03711-f009:**
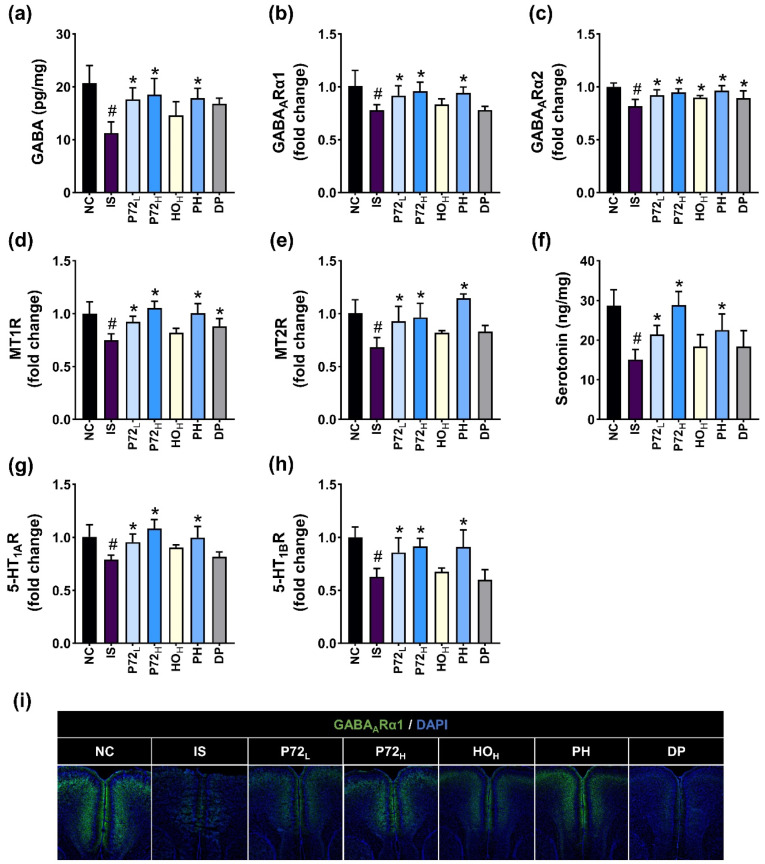
Effects of P72, HO, and PH on IS-suppressed DA- and insomnia-related biomarker levels in the prefrontal cortex. Their effects on GABA (**a**), GABA_A_Rα1 (**b**), and GABA_A_Rα2 (**c**) levels. Their effects on MT1R (**d**) and MT2R (**e**) levels. Their effects on serotonin (**f**), 5-HT_1A_R (**g**), and 5-HT_1B_R (**h**) levels. (**i**) Their effects on GABA_A_Rα1-positive cell populations. IS, vehicle; P72_L_, 4 × 10^8^ CFU/mL of P72; P72_H_, 1 × 10^9^ CFU/mouse of P72; HO_H_, 0.24 g/kg of HO; PH, 0.4 × 10^9^ CFU/mouse/day of P72 and 0.12 g/kg of HO; DP, 20 mg/kg DP, in IS/PS-exposed mice; NC, vehicle, in IS/PS-nontreated mice. Data are mean ± SD (*n* = 6). ^#^ *p* < 0.05 vs. NC. * *p* < 0.05 vs. IS.

**Figure 10 nutrients-16-03711-f010:**
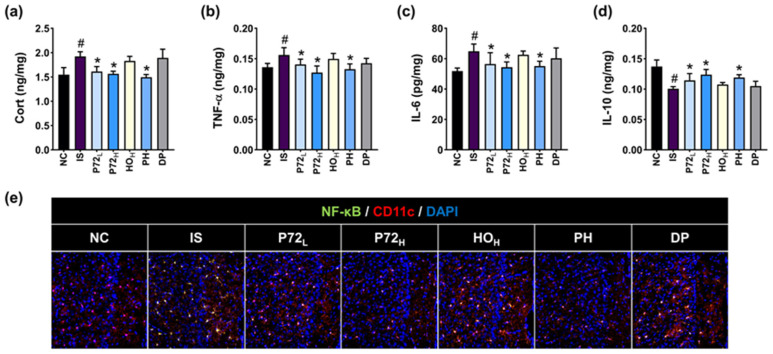
Effects of P72, HO, and PH on IS-induced inflammatory biomarker levels in the prefrontal cortex. Their effects on Cort (**a**), TNF-α (**b**), IL-6 (**c**), and IL-10 (**d**) levels and NF-κB^+^ Iba1^+^ cell numbers (**e**). IS, vehicle; P72_L_, 4 × 10^8^ CFU/mL of P72; P72_H_, 1 × 10^9^ CFU/mouse of P72; HO_H_, 0.24 g/kg of HO; PH, 0.4 × 10^9^ CFU/mouse/day of P72 and 0.12 g/kg of HO; DP, 20 mg/kg DP, in IS/PS-exposed mice; NC, vehicle, in IS/PS-nontreated mice. Data are mean ± SD (*n* = 8). ^#^ *p* < 0.05 vs. NC. * *p* < 0.05 vs. IS.

**Figure 11 nutrients-16-03711-f011:**
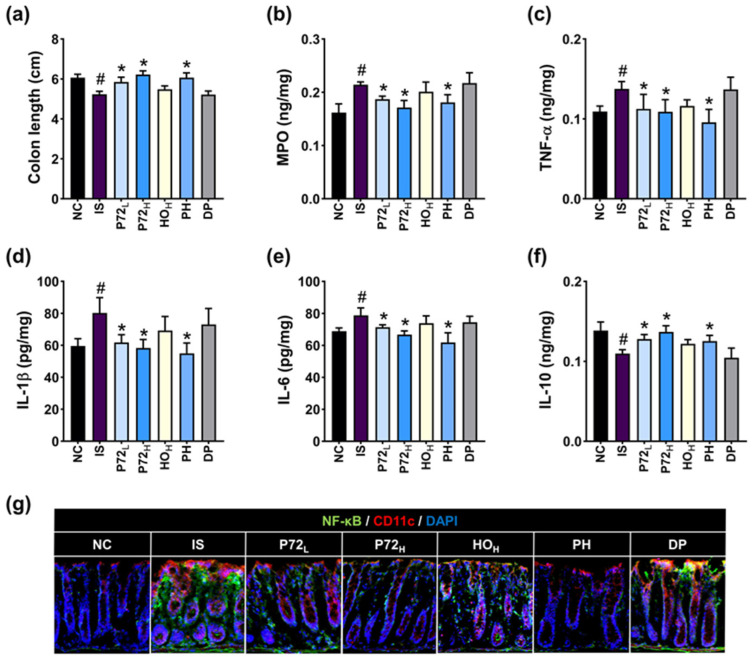
Effects of P72, HO, and PH on IS-induced colitis in mice. Their effects on colon length (**a**), myeloperoxidase (MPO, (**b**)), TNF-α (**c**), IL-1β (**d**), IL-6 (**e**), and IL-10 expression (**f**), and NF-κB^+^ CD11c^+^ cell populations (**g**) in the colon. IS, vehicle; P72_L_, 4 × 10^8^ CFU/mL of P72; P72_H_, 1 × 10^9^ CFU/mouse of P72; HO_H_, 0.24 g/kg of HO; PH, 0.4 × 10^9^ CFU/mouse/day of P72 and 0.12 g/kg of HO; DP, 20 mg/kg DP, in IS/PS-exposed mice; NC, vehicle, in IS/PS-nontreated mice. Data are mean ± SD (*n* = 8). ^#^ *p* < 0.05 vs. NC. * *p* < 0.05 vs. IS.

**Figure 12 nutrients-16-03711-f012:**
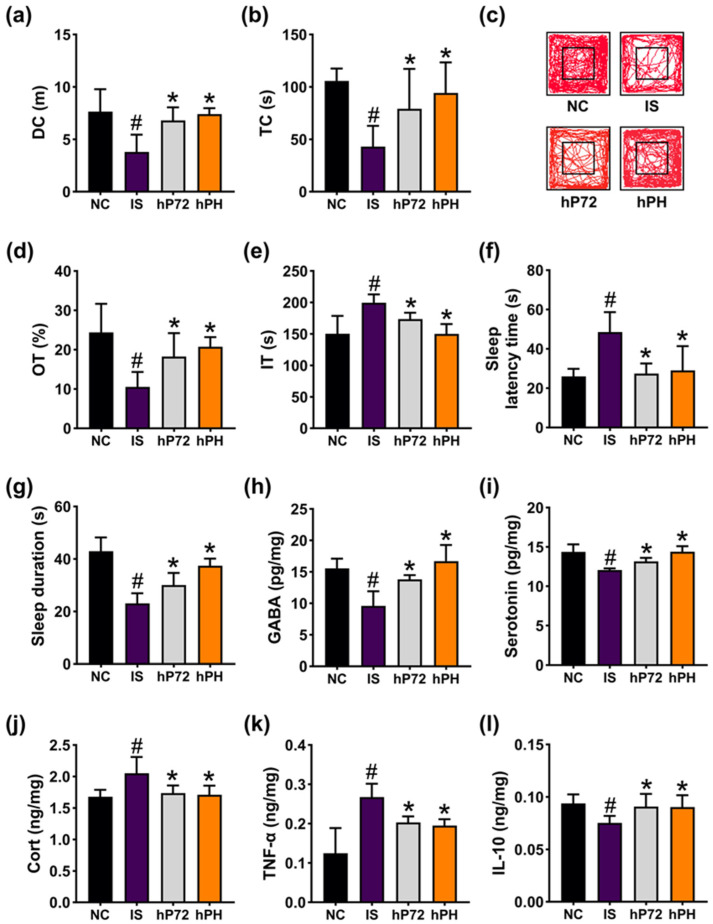
Effects of hP72 and hPH on DI_L_S in IS-treated mice. Their effects on DC (**a**), TC (**b**), and travel pathway (**c**) in OFT. Their effects on OT (**d**) in EPMT and IT in TST (**e**). Their effects on SLT (**f**) and SLD (**g**). Their effects on GABA (**h**), serotonin (**i**), Cort (**j**), TNF-α (**k**), and IL-10 (**l**) levels in the prefrontal cortex. IS, vehicle; hP72, 1 × 10^9^ CFU/mouse of hP72; hPH, 1 × 10^9^ CFU/mouse of hP72 and 0.12 g/kg of HO in IS/IF-treated mice; NC, IS/IF-nontreated mice. Data are mean ± SD (*n* = 8). ^#^ *p* < 0.05 vs. NC. * *p* < 0.05 vs. IS.

## Data Availability

The datasets generated during the current study are available from the corresponding author on reasonable request.

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
