# Peer review of "Alleviation of Immobilization Stress or Fecal Microbiota-Induced Insomnia and Depression-like Behaviors in Mice by Lactobacillus plantarum and Its Supplement"

_nutrients, 2024, doi:10.3390/nu16213711_

Round 1
Reviewer 1 Report
Comments and Suggestions for Authors
The authors evaluated the beneficial effect of Lactobacillus plantarum P72 (
both live and heat-killed P72 (hP72)) and its supplement in mice with DA- and SL-like symptom (DSLS). These results show that P72 and its combination with HO may alleviate DA and SL by upregulating serotonergic and GABAergic systems through the suppression of NF-κB activation. However, I would like to raise several major concerns as follows.
1. Stress causes gut dysbiosis. The title mentions gut bacteria-induced sleeplessness. However, in this paper, there is no evidence of intestinal microbial sequencing, whether the intestinal flora changes after the intervention of Lactobacillus plantarum P72 and its supplement.
2. Why use IS-induced colitis in mice, and detect some factors related to inflammation? These inflammation factors in gut can be detected in IS-induced DA- and sleep disturbance mice.
3. In line 92, 2×108 CFU/mouse, What medium was used for the culture count, or how was it counted?
4. There are grammatical errors in the manuscript, which need to be checked. Like in line 97, the sentence “primers is indicated” should be revised as “primers are indicated”.
Comments on the Quality of English Language4. There are grammatical errors in the manuscript, which need to be checked. Like in line 97, the sentence “primers is indicated” should be revised as “primers are indicated”.
Author Response
Comments and Suggestions for Authors
The authors evaluated the beneficial effect of Lactobacillus plantarum P72 (both live and heat-killed P72 (hP72)) and its supplement in mice with DA- and SL-like symptom (DSLS). These results show that P72 and its combination with HO may alleviate DA and SL by upregulating serotonergic and GABAergic systems through the suppression of NF-κB activation. However, I would like to raise several major concerns as follows.
- Stress causes gut dysbiosis. The title mentions gut bacteria-induced sleeplessness. However, in this paper, there is no evidence of intestinal microbial sequencing, whether the intestinal flora changes after the intervention of Lactobacillus plantarum P72 and its supplement.
--> Thank you for your comment. We prepared experimental mice with depression and insomnia by transplanting gut microbiota, as previous report (Reference 25).
- Why use IS-induced colitis in mice, and detect some factors related to inflammation? These inflammation factors in gut can be detected in IS-induced DA- and sleep disturbance mice.
--> Thank you. IS treatment simultaneously caused DA, sleep disturbance, and colitis in mice. Therefore, we examined whether P72 treatment could alleviate DA, sleep disturbance, and colitis.
- In line 92, 2×108 CFU/mouse, What medium was used for the culture count, or how was it counted?
--> Thank you for your comment. We described it in Lines 59-61.
- There are grammatical errors in the manuscript, which need to be checked. Like in line 97, the sentence “primers is indicated” should be revised as “primers are indicated”.
--> Thank you. We revised your suggested and other errors.
Reviewer 2 Report
Comments and Suggestions for Authors
Comments and Suggestions for Authors: This article provides a comprehensive introduction to how Lactobacillus Plantarum and its supplements effectively alleviate immobilization stress or gut bacteria-Induced sleeplessness and depression-like symptoms by activating Nf-κB to regulate the serotonergic and gabaergic systems.
The background section introduces the beneficial activities exhibited by various probiotics on mental disorders, thereby leading to the introduction of the source of raw materials for the main experiment on P72, providing readers with a more relevant reading experience. The experimental design is comprehensive and reasonable, and this study has included both animal experiments and cell experiments to investigate the phenomenon from both macroscopic and molecular perspectives. The experimental data analysis in the Results section is accurate, the experimental results are reliable, and the figures and tables are drawn in a standardized manner, so that readers can understand the research content simply by reading the figures. Additionally, the presentation of the experimental results is progressive, revealing the results of each aspect in a sequential manner, thus gradually deducing the conclusion. In the discussion section, the authors provide a mechanistic explanation and delve into how P72 affects mouse behavior and physiological changes by upregulating the 5-HT and GABA systems while inhibiting NF-κB activation. Additionally, they also discuss the comparison of this study with similar ones. In general, this article is innovative in terms of research content, reasonable in experimental design, and well-written. The data results are reliable, making it a highly readable paper.
Author Response
Comments and Suggestions for Authors: This article provides a comprehensive introduction to how Lactobacillus Plantarum and its supplements effectively alleviate immobilization stress or gut bacteria-Induced sleeplessness and depression-like symptoms by activating Nf-κB to regulate the serotonergic and gabaergic systems.
The background section introduces the beneficial activities exhibited by various probiotics on mental disorders, thereby leading to the introduction of the source of raw materials for the main experiment on P72, providing readers with a more relevant reading experience. The experimental design is comprehensive and reasonable, and this study has included both animal experiments and cell experiments to investigate the phenomenon from both macroscopic and molecular perspectives. The experimental data analysis in the Results section is accurate, the experimental results are reliable, and the figures and tables are drawn in a standardized manner, so that readers can understand the research content simply by reading the figures. Additionally, the presentation of the experimental results is progressive, revealing the results of each aspect in a sequential manner, thus gradually deducing the conclusion. In the discussion section, the authors provide a mechanistic explanation and delve into how P72 affects mouse behavior and physiological changes by upregulating the 5-HT and GABA systems while inhibiting NF-κB activation. Additionally, they also discuss the comparison of this study with similar ones. In general, this article is innovative in terms of research content, reasonable in experimental design, and well-written. The data results are reliable, making it a highly readable paper.
--> Thank you for your comment. We revised our manuscript according to your suggestions.
Reviewer 3 Report
Comments and Suggestions for Authors
In the present manuscript Dong-Yun Lee et colleagues aim to evaluate the effects of lactobacillus (Lactiplantibacillus) plantarum P72 on depression/anxiety and insomnia in mice exposed to immobilization stress or cultured fecal microbiota. While the potential of using a probiotic to tread depression and insomnia symptoms is certainly valuable, the data presented are quite difficult to follow. The abbreviations used are a bit confusing and sometimes misleading (i.e. P72 is listed as P72 or LP), other times are not specified at all. Figures are confusing and not properly discussed. It is quite difficult to follow the order of the data presented because figures letter are not listed in the manuscript and authors sometimes discuss the last part of the figure and then come back to the initial part.
Fig. 1: This reviewer assume that NC is negative control but it should specified into the manuscript. No information of figure 1a and b is given in the main text. On line 143 authors state that several lactobacilli have been tested and P72 is the most relevant. However, these data are not shown in the manuscript. Additionally what n=4 refers to? To the number of in vitro tests done or to the number of biological replicates?
Fig. 2: It is quite difficult to follow it. There are no indication in the text of fig. 2 a to i. What is F2,21?. Result section of Fig 2 are not clear. First it is discussed the effect of IS in all the figures, then of the P72 not following any order. Fig 2 d is discussed when the authors already discussed Fig. e-i. Assuming authors are referring to fig 2a, it´s quite difficult to believe that IS decrease TD of 85.3% of the NC. Same for the fig 2g (?) where authors state that the increase of IS is the 142.8% of the NC.
Fig. 3: Again, there is no indication in the text about Fig 3a to i. Additionally in the results section of Fig 3 authors jump from figure 3a to figure 3 h-i and then they go back to fig. 3a, making the whole part very difficult to follow.
Fig. 4: No indication in the text about Fig 4a to f. Additionally the results are presented no following the order of the figure.
Fig. 5: Same issues of the Fig. 2, including the Fig 2,21 problem.
Fig. 6: Same issues of the Fig. 4. What is Fig 6,49?
Fig. 7: No indication of Fig. 7a to f has been given into the text. The abbreviation of myeloperoxidase in MPO is not mention. Lines 235-237 are wrongly aligned.
Fig. 8: same issues of all the other figures (no information about 8a to I, discussed not in order of the presented graph). What is F6,49? Assuming authors are referring to fig 8a, it´s quite difficult to believe that IS decrease TD of 88.8% of the NC Additionally, the abbreviations are quite difficult to follow, a table under each graph with all the treatment listed would probably benefit the clarity of the figure.
Fig. 9-10: No information of the figure 9 a to I and Figure 10a to e. Results section describing this to figures is really poor.
Fig. 11: No information of Figure 11a to g. Moreover, Figure11g seems to be discussed before than a, b, c, d, e and f.
Fig. 12: No information of Figure 12a to l.
Conclusions are confused and do not follow the order as the manuscript is presented.
Comments on the Quality of English Languagenone
Author Response
Comments and Suggestions for Authors
In the present manuscript Dong-Yun Lee et colleagues aim to evaluate the effects of lactobacillus (Lactiplantibacillus) plantarum P72 on depression/anxiety and insomnia in mice exposed to immobilization stress or cultured fecal microbiota. While the potential of using a probiotic to tread depression and insomnia symptoms is certainly valuable, the data presented are quite difficult to follow. The abbreviations used are a bit confusing and sometimes misleading (i.e. P72 is listed as P72 or LP), other times are not specified at all. Figures are confusing and not properly discussed. It is quite difficult to follow the order of the data presented because figures letter are not listed in the manuscript and authors sometimes discuss the last part of the figure and then come back to the initial part.
Fig. 1: This reviewer assume that NC is negative control but it should specified into the manuscript. No information of figure 1a and b is given in the main text. On line 143 authors state that several lactobacilli have been tested and P72 is the most relevant. However, these data are not shown in the manuscript. Additionally what n=4 refers to? To the number of in vitro tests done or to the number of biological replicates?
--> Thank you for your comment. We tested the serotonin-inducing and anti-inflammatory activities for 25 lactobacilli isolated from healthy human feces. P72 most potently induced serotonin secretion from SH-SY5Y cells. And n=4 means that we tested four replicates and measured serotonin. (line 157).
Fig. 2: It is quite difficult to follow it. There are no indication in the text of fig. 2 a to i. What is F2,21?. Result section of Fig 2 are not clear. First it is discussed the effect of IS in all the figures, then of the P72 not following any order. Fig 2 d is discussed when the authors already discussed Fig. e-i. Assuming authors are referring to fig 2a, it´s quite difficult to believe that IS decrease TD of 85.3% of the NC. Same for the fig 2g (?) where authors state that the increase of IS is the 142.8% of the NC.
--> Thank you. We drew the Figure 2 and revised its result description according to your comments (Line 159-171). We indicated F-ratio and degrees of freedom and P values in the sentence to understand the sizes of groups their animal numbers the distribution of results, and significance between groups. Although the animal experimental data were variously compared, the discussion was confused. Therefore, we simply compared all data of experimental animals to those NC group.
Fig. 3: Again, there is no indication in the text about Fig 3a to i. Additionally in the results section of Fig 3 authors jump from figure 3a to figure 3 h-i and then they go back to fig. 3a, making the whole part very difficult to follow.
--> Thank you. We drew the Figure 3 and revised its result description according to your comments (Line 177-183).
Fig. 4: No indication in the text about Fig 4a to f. Additionally the results are presented no following the order of the figure.
--> Thank you. We drew the Figure 4 and revised its result description according to your comments (Line 191-194).
Fig. 5: Same issues of the Fig. 2, including the Fig 2,21 problem.
--> Thank you. We drew the Figure 5 and revised its result description according to your comments (Line 204-212).
Fig. 6: Same issues of the Fig. 4. What is Fig 6,49?
--> Thank you. We drew the Figure 6 and revised its result description according to your comments (Line 218-224). We indicated F-ratio and degrees of freedom and P values in the sentence to understand the sizes of groups their animal numbers the distribution of results, and significance between groups. Although the animal experimental data were variously compared, the discussion was confused.
Fig. 7: No indication of Fig. 7a to f has been given into the text. The abbreviation of myeloperoxidase in MPO is not mention. Lines 235-237 are wrongly aligned.
--> Thank you. We drew the Figure 7 and revised its result description according to your comments (Line 230-238).
Fig. 8: same issues of all the other figures (no information about 8a to I, discussed not in order of the presented graph). What is F6,49? Assuming authors are referring to fig 8a, it´s quite difficult to believe that IS decrease TD of 88.8% of the NC Additionally, the abbreviations are quite difficult to follow, a table under each graph with all the treatment listed would probably benefit the clarity of the figure.
--> Thank you. We drew the Figure 8 and revised its result description according to your comments (Line 243-259). Although the animal experimental data were variously compared, the discussion was confused. Therefore, we simply compared all data of experimental animals to those NC group.
Fig. 9-10: No information of the figure 9 a to I and Figure 10a to e. Results section describing this to figures is really poor.
--> Thank you. We drew the Figures 9 and 10 and revised its result description according to your comments (Line 266-282).
Fig. 11: No information of Figure 11a to g. Moreover, Figure11g seems to be discussed before than a, b, c, d, e and f.
--> Thank you. We drew the Figure 11 and revised its result description according to your comments (Line 288-293).
Fig. 12: No information of Figure 12a to l.
--> Thank you. We drew the Figure 12 and revised its result description according to your comments (Line 301-313).
Conclusions are confused and do not follow the order as the manuscript is presented.
--> Thank you. We revised Conclusion section according to your comment.
Round 2
Reviewer 1 Report
Comments and Suggestions for Authors
1. The title of manuscript in the supplementary material is inconsistent with the title in the text.
2. In line 60, the precipitate of P72 was freeze-dried. There was a protective agent or vector for freeze-dried condition of P72, but there was no placebo group in animal trials as the control group.
3. Was P72 inactivated after freeze-drying?
4. In line 90, “The cultured fecal microbiota (2×108 colony-forming unit [CFU]/mouse, suspended in 0.1 90 mL of saline) of patients with depression and inflammatory bowel disease (DI)”. The author's reply was that the count of the cultured fecal microbiota using MRS Medium, which is a selective medium for lactic acid bacteria. 2×108 colony-forming unit is the viable number of lactic acid bacteria?
5.The concentrations of LPS ( 100 ng/mL) and corticosterone (300 μM) were used for Caco-2 cells and SH-SY5Y cells, respectively. However, Whether the use of this concentration is the optimal action concentration found by the author, or a reference, the article is not stated.
6. Use the doses of P72 at 0.4*109 and 1.0*109 CFU/mL, how do you think about these two doses, the difference in dose is too small.
7. In the manuscript, PH, 0.4×109 CFU/mouse/day of P72 and 0.12 g/kg of HO, hPH, 1×109 CFU/mouse/day of hP72 and 0.12 g/kg of HO, How to consider the different doses used for P72.
Comments on the Quality of English Language1. in line 127, delete 00.
2. in line 221, "and IL-10 Orally administred P72 upregulated cFM transplantation-suppressed GABA", IL-10?
3. The grammar of the whole manuscript has to be checked carefully.
Author Response
We revised our manuscript according to the suggestions of your comments. The revised sentences in the manuscript are yellow-highlighted.
Comments and Suggestions for Authors
- The title of manuscript in the supplementary material is inconsistent with the title in the text.
-->Thank you. We revised it.
- In line 60, the precipitate of P72 was freeze-dried. There was a protective agent or vector for freeze-dried condition of P72, but there was no placebo group in animal trials as the control group.
--> We described it at detail in Lines 60-61.
- Was P72 inactivated after freeze-drying?
-->Thank you for your comment. We described it in Lines 63-64.
- In line 90, “The cultured fecal microbiota (2×108 colony-forming unit [CFU]/mouse, suspended in 0.1 90 mL of saline) of patients with depression and inflammatory bowel disease (DI)”. The author's reply was that the count of the cultured fecal microbiota using MRS Medium, which is a selective medium for lactic acid bacteria. 2×108 colony-forming unit is the viable number of lactic acid bacteria?
--> Thank you for your comment. Number of cultured fecal microbiota were counted by using general anaerobic medium [Nissui Pharm. Co., Tokyo, Japan] MRS was used for the count of Lactobacilli.
5.The concentrations of LPS ( 100 ng/mL) and corticosterone (300 μM) were used for Caco-2 cells and SH-SY5Y cells, respectively. However, Whether the use of this concentration is the optimal action concentration found by the author, or a reference, the article is not stated.
-->Thank you. We cited it.
- Use the doses of P72 at0.4*109 and 1.0*109 CFU/mL, how do you think about these two doses, the difference in dose is too small.
-->Thank you. We decided the dose of P72 through the preliminary experiment.
- In the manuscript, PH, 0.4×109 CFU/mouse/day of P72 and 0.12 g/kg of HO, hPH, 1×109 CFU/mouse/day of hP72 and 0.12 g/kg of HO, How to consider the different doses used for P72.
-->Thank you. To understand the combined effect of P72 and HO, we decided the doses of P72and H, based on the preliminary experiment of P72 and the commonly used dose of HO in human (12-fold in mice), respectively.
Comments on the Quality of English Language
- in line 127, delete 00.
-->Thank you. We deleted it.
- in line 221, "and IL-10 Orally administred P72 upregulated cFM transplantation-suppressed GABA", IL-10?
-->Thank you. We revised it.
- The grammar of the whole manuscript has to be checked carefully.
-->Thank you. We checked and revised our manuscript according to your comments.
Reviewer 3 Report
Comments and Suggestions for Authors
Dong-Yun Lee et colleagues have improved the whole manuscript. However, some info are still missing.
Lines 143-146: Authors wrote “we screened serotonin expression-inducing probiotics from human fecal bacteria collection using corticosterone-stimulated SH-SY5Y cells. Of tested lactobacilli, P72 the most potently increased corticosterone-suppressed serotonin production (Figure 1a)”. However, fig 1a only show the serotonin production of P72. The comparison between P72 and all the other lactobacilli tested is still missing.
When appropriate authors should explain somewhere in the manuscript (perhaps in material and methods section?) what is the F-ratio and degrees of freedom. This explanation is missing in the text.
All figure legends are wrongly formatted.
Author Response
We revised our manuscript according to the suggestions of your comments. The revised sentences in the manuscript are yellow-highlighted.
Dong-Yun Lee et colleagues have improved the whole manuscript. However, some info are still missing.
Lines 143-146: Authors wrote “we screened serotonin expression-inducing probiotics from human fecal bacteria collection using corticosterone-stimulated SH-SY5Y cells. Of tested lactobacilli, P72 the most potently increased corticosterone-suppressed serotonin production (Figure 1a)”. However, fig 1a only show the serotonin production of P72. The comparison between P72 and all the other lactobacilli tested is still missing.
-->Thank you. We described the data of the most potent P72 due to a lot of data (including P71, Lactobacillus casei; P72, Lactobacillus plantarum, P73, L plantarum; P73, L. pentosus; P75, L. gasseri, …..).
When appropriate authors should explain somewhere in the manuscript (perhaps in material and methods section?) what is the F-ratio and degrees of freedom. This explanation is missing in the text.
-->Thank you. We added it in Line 140-141.
All figure legends are wrongly formatted.
-->We checked and revised them again.